# Migraine Management in Community Pharmacies: Knowledge, Attitude and Practice Patterns of Pharmacists in Saudi Arabia

**DOI:** 10.3390/pharmacy11050155

**Published:** 2023-09-24

**Authors:** Fahad Alzahrani, Yaser M. Alahmadi, Sultan S. Al Thagfan, Sultan Alolayan, Hossein M. Elbadawy

**Affiliations:** 1Department of Clinical and Hospital Pharmacy, College of Pharmacy, Taibah University, Madinah 42353, Saudi Arabia; yahmadi@taibahu.edu.sa (Y.M.A.); salthagfan@gmail.com (S.S.A.T.); solayan@taibahu.edu.sa (S.A.); 2Department of Pharmacology and Toxicology, College of Pharmacy, Taibah University, Madinah 42353, Saudi Arabia; hmbadawy@taibahu.edu.sa

**Keywords:** attitude, community pharmacists, pharmacy practice, knowledge, migraine, practice, Saudi Arabia

## Abstract

In Saudi Arabia, community pharmacies offer healthcare services for different conditions. However, clarity of the competence of pharmacists in managing migraines is lacking. This study aimed to explore the current knowledge, attitude, and practice patterns of community pharmacists concerning migraine management in the northwestern part of Saudi Arabia. A cross-sectional study was carried out between June and September 2022 among 215 Saudi community pharmacists. Data analysis was performed by descriptive and inferential statistics using SPSS version 27. Most community pharmacists (87.9%) feel that migraine management is essential to their practice, and 83.3% suggest between one and five over-the-counter (OTC) migraine products daily. Among the study pharmacists, 83.7% feel migraine patients should try OTC before prescription medications. Only 9.3% of the community pharmacists do not believe that migraine is a neurological disorder. The medications most prescribed for migraine were triptans, representing 52.1% of prescriptions. There were significant differences between the gender of the pharmacists and their knowledge, attitude, and practice overall score (*p*-value = 0.04). Male pharmacists exhibited higher knowledge, attitude, and practice scores than female pharmacists. Although many community pharmacists acknowledge their expertise and involvement in managing migraines, there is a requirement for further education and training to enhance their capacity to offer complete care to migraine patients. Pharmacists should also consider non-pharmacological interventions and complementary therapies when treating migraine symptoms.

## 1. Introduction

Migraines are a lifelong neurological condition that causes primary headaches and contributes to global disability [1]. The exact cause of a migraine is not fully understood; however, it is believed to involve a combination of genetic, environmental, and neurological factors [2]. According to the World Health Organization (WHO), migraines are the third-most prevalent neurological disorder and second-most disabling disorder globally [3,4]. It is estimated to affect approximately one billion people worldwide, making it one of the most common neurological disorders. Women are more prone to migraines than men are, with hormonal fluctuations often triggering or worsening them [1]. Migraine onset can occur at any point in life; however, the prevalence tends to be highest before age 45 [5].

In Arab countries, several studies have assessed migraine prevalence. In Saudi Arabia, approximately 25% of the population experiences migraines within one year [6]. Similarly, a recent investigation in Kuwait revealed that 23% of residents in the country suffer from migraines [7]. Another study in Egypt focused on migraine prevalence in the Fayoum Governorate, reporting a 17.3% rate over one year [8]. These findings highlight the significant impact of migraines on individuals across various Arab countries.

People who suffer from migraines frequently rely on over-the-counter (OTC) medications for alleviation, yet these remedies might offer only partial efficacy when managing intense episodes. Wenzel et al. stated that individuals with disabilities during migraine attacks are poor candidates for OTC medication and should consult a physician for prescription migraine drugs [9]. An earlier research article indicated that 49% of patients used OTC medications exclusively, whereas 29% intermittently combined OTC with prescription medications [10]. Over-reliance on “painkiller” medications is a prevalent issue with far-reaching consequences, particularly in migraines. This over-reliance can inadvertently lead to chronic daily headaches (CDH), a condition afflicting a staggering number of individuals daily. Approximately ten million people are estimated to be affected by CDH worldwide. Effective patient education and monitoring play a crucial role in preventing medication misuse. Healthcare providers can significantly reduce the chances of medication misuse by providing patients with essential information and closely monitoring their medication use [11,12,13].

In collaboration with the WHO, the European Headache Federation (EHF) has published several migraine management guidelines [14]. According to this guideline, a mild migraine should be treated with an analgesic or NSAID. When this first step fails, triptans are indicated. Nevertheless, triptans often necessitate a prescription due to their potential adverse effects and the possibility of interactions with other medications [15]. Additionally, many countries, including Saudi Arabia, still use ergotamine for mild to moderate acute migraines [16]. Despite the availability of migraine treatment guidelines, physicians often treat migraine patients inappropriately [17,18,19]. Gupta et al. reported that many migraine patients leave the hospital without receiving an outpatient diagnosis, medications, or physician instructions due to the lack of time to manage the complexities of chronic pain, including migraine headaches [18].

Pharmacists are crucial in supporting migraine patients due to their accessibility and expertise in healthcare. Patients often seek their advice, making them well-placed to assist [20]. Patel et al. stated that pharmacists possess valuable information, which allows them to make informed decisions and better guide patients regarding migraines. More specifically, they can explain the reasons behind preventive therapy and provide insights on how and when it should be initiated [21]. According to a Wenzel and his colleagues’ survey, -seven percent of community pharmacists were found to recommend at least one OTC daily for headache treatment [22].

Moreover, the latest OTC survey by Pharmacy Times estimated that pharmacists in the United States make nearly 1.9 million OTC recommendations for headaches monthly. Of these recommendations, approximately 800,000 are specifically for migraine treatment [23]. These statistics highlight pharmacists’ significant roles in providing OTC recommendations for headache relief, particularly in the case of migraines.

In Saudi Arabia, community pharmacies are regulated by The Ministry of Health (MOH) and the Saudi Food and Drug Authority (SFDA). The SFDA ensures the safety, quality, and effectiveness of pharmaceutical goods within the nation, whereas the MOH oversees the healthcare system [24]. Local pharmacies are authorized to provide prescription drugs prescribed by licensed medical practitioners, except for those classified as non-prescription (OTC) as defined by regulations. Certain specialized medications or therapies might exclusively be accessible at particular healthcare establishments via specialized clinics [25].

There are several opportunities to improve headache care. While community pharmacists are well-positioned to enhance migraine management, little is known about their competence in various complaints related to headaches. A study in Madinah, Saudi Arabia, examined how pharmacists manage headaches and analyzed their knowledge, attitude, and practice toward the condition. However, there were some limitations to the study. Firstly, it was only conducted in one city and only encompassed part of the western region. Secondly, the survey questions were focused on regular headaches rather than migraines. Lastly, the study did not solely focus on community pharmacists but included data from community and hospital pharmacists [26]. Therefore, this study aimed to evaluate community pharmacists’ present state of knowledge, attitudes, and current practice regarding migraine therapy in the northwestern part of Saudi Arabia. Analyzing the knowledge, attitude, and present practice of local pharmacists in Saudi Arabia concerning the management of migraines can offer healthcare systems valuable comprehension of the particular difficulties and prospects in the nation. This understanding can pave the way for focused measures, protocols, and educational campaigns that augment community pharmacists’ contributions and enhance the comprehensive control of migraines within the Saudi Arabian population.

## 2. Materials and Methods

### 2.1. Study Design and Setting

A cross-sectional observational study was conducted among pharmacists in the northwestern part (Madinah and Tabuk regions) of Saudi Arabia between June and September 2022. A community pharmacist with a bachelor’s degree in pharmacy, a higher education degree, and a Saudi Arabian pharmacy license were eligible for this study. A pharmacist who could not consent to participate or was a trained pharmacist during the study was excluded.

### 2.2. Sample Size and Sampling Procedure

Based on a sample size calculator (www.raosoft.com, accessed on 18 July 2023), the sample size was calculated at a 95% confidence interval (CI), accepting a margin of error of 5% for about 788 pharmacists practicing in community pharmacies in the northwestern part of Saudi Arabia [27]. This study was estimated to have a sample size of 259 pharmacists. A final analysis was performed on 215 pharmacists after the study period had ended. During data collection, 44 community pharmacists were excluded because of their incomplete survey responses.

The survey was conducted using an online Google form. Community pharmacists were recruited through various methods, including social media campaigns on Twitter and WhatsApp groups. Furthermore, several national pharmacy chains advertised the study through internal distribution lists. Besides reaching out to urban pharmacies, the online survey was also shared with regional and rural pharmacies. Regardless of how the questionnaire was obtained, participants were asked to complete it only once.

### 2.3. Data Collection Form

The investigators developed a modified version of previously used surveys to evaluate ‘general practitioners’ knowledge, attitude, and practice patterns regarding headaches, which has been transformed into a structured, self-administered pharmacist’s migraine survey (PMS) [28]. The survey used for general practitioners was created to adhere to the migraine treatment guidelines of the United States Headache Consortium (the “consortium”). Three faculty members at the College of Pharmacy, Taibah University, evaluated the survey’s face and content validity. The study consisted of 27 questions about the sample’s demographics and professional characteristics. Pharmacists were provided with statements and asked to select from “strongly disagree”, “disagree”, “neutral”, “agree”, or “strongly agree” to evaluate their knowledge, attitudes, and current practices regarding migraine management.

### 2.4. Validity and Reliability of the Study Tool

The relevance and clarity of the survey questions were evaluated through a pilot study with eight pharmacists to ensure readability and comprehensibility. The pharmacists who first reviewed the questionnaire shared their thoughts on each question. Certain statements were adjusted or reformulated according to the feedback provided by the participants. Data from the pilot sample was excluded from the final one.

The consistency of scores was examined through the test-retest technique, evaluating it across a brief time span. To gauge stability, twelve pharmacy alumni who were not part of the study were requested to complete the questionnaire twice, with a short time gap between the two instances. The scores from the same participants in both iterations were compared, and Pearson’s correlation coefficient (r) was employed to indicate the reliability of the test-retest. It was decided a priori that a more than 80% correlation coefficient would be needed to ensure score stability over the short time between the two rounds. Pearson’s correlation coefficient was 0.96 (95% CI of 0.91 to 0.98) with a *p*-value of <0.01, indicating excellent score stability [29]. Cronbach’s α determined internal consistency. Internally consistent tools should have 0.70 ≤ α ≤ 0.95 [30,31]. When all items were included, Cronbach’s α was 0.81. When Cronbach’s α was computed for each domain separately, the knowledge items had a Cronbach’s alpha of 0.76, attitude items of 0.78, and the use of protective measures of 0.89, which indicated that the items were internally consistent across all domains.

### 2.5. Statistical Analysis

Data from community pharmacists were entered into and analyzed using the SPSS version 27 for Windows. Descriptive statistics, including frequency and percentage were used to summarize the data. The overall score by community pharmacists in the survey sections was developed for the study. Regarding the queries concerning knowledge, attitude, and practice, a numerical value was assigned to each response, with “strongly disagree” corresponding to a value of one and “strongly agree” to a value of five. An average score was then computed by aggregating the individual scores from the questions. Greater scores are indicative of increased knowledge, a favorable attitude, and improved practice. The Shapiro-Wilk tests assessed the scores for normality distribution. Without normal distribution, the Mann-Whitney-U or Kruskal-Wallis tests were used to compare differences as appropriate. The effect size is small if the r value varies around 0.1, medium if the r varies around 0.3, and large if the r varies more than 0.5. A two-tailed *p*-value < 0.05 was considered statistically significant.

## 3. Results

### 3.1. Characteristics of Study Participants

Out of 788 community pharmacists in the northwestern part of Saudi Arabia, only 215 participated in our study, resulting in a response rate of 27.28%. Males represented 89.8% of the participants, while 10.2% were females. The median age in the study was 31.0, and the interquartile range (IQR) was 28–36. Regarding the highest qualification for each participant, 85.6% of pharmacists had a bachelor’s degree, 11.2% a PharmD, 2.3% a master’s degree, and 0.9% of the participating pharmacists had a doctorate (PhD).

Regarding the years of experience as practicing pharmacists, 36.7% had more than ten years of experience practicing pharmacy, while 1.4% had less than one year. Most pharmacists (76.3%) graduated from non-Saudi universities, while 23.7% were local graduates obtaining their pharmacy degrees from a Saudi university. On average, 96.7% of participants suggested daily OTC medications to their patients.

Community pharmacists can manage various types of headaches. Pharmacists reported that sinus headaches accounted for 28.3% of headaches, followed by migraines (26.5%) and tension (20.9%). Regarding continuous education on migraine or headache management, more than half of the pharmacists (54.9%) reported attending a course or educational event about headache/migraine management. All pharmacists referred at least one patient to a physician every month, primarily on suspicion of chronic illness (45.2%) or inadequate treatment (28.8%). In 41.4% of responses, less than five migraine patients visited the pharmacy monthly. While in 9.8% of pharmacies, 11–15 migraine patients visited their pharmacy monthly (Table 1).

### 3.2. Knowledge, Attitude, and Current Practice of Pharmacists Regarding Migraine Management

Sixteen questions measured pharmacists’ knowledge, attitude, and practice regarding migraine. Table 2 summarizes community pharmacists’ responses to survey statements, divided into strongly disagree, disagree, neutral, agree, and strongly agree categories. When asked whether headache patients should try OTC drugs first and then try prescription drugs, 83.7% agreed or strongly agreed. Before advising an OTC drug, 86.5% agreed or strongly agreed that they always ask the patient about headache attack-related disability, while 12.6% were neutral. Pharmacists were asked if they teach patients to guard against OTC drug overuse, and 82.8% agreed or strongly agreed.

Regarding teaching migraine sufferers to guard against prescription medication overuse, 76.7% agreed or strongly agreed, and 18.6% were neutral. When deciding on drug choice, merely 5.6% of the respondents expressed disagreement or strong disagreement with the notion that migraine-specific drugs (triptans) should be used for patients who haven’t responded well to at least two other prescription medications. Regarding the statement, “Migraine is primarily a disease of the brain, with a well-established neurological basis”, 74.4% concurred or strongly concurred, while 16.3% maintained a neutral stance. Most community pharmacists agreed or strongly agreed that headache patients were essential to their practice. Among all community pharmacists, 64.6% agreed or strongly agreed concerning their ability to identify a patient needing preventive migraine medication. 79.5% of the pharmacists responded, agreed, or strongly agreed to always encourage a patient to maintain a headache diary. Agreed or strongly agreed was reported by 75.3% regarding always discussing nonmedication treatments for headaches, while 17.2% were neutral. The fact that patient satisfaction is an essential consideration in headache treatment was strongly agreed by 44.2% and agreed by 43.7%.

Knowing when to refer a patient to a physician was agreed upon by 69.7%, neutral by 25.6%, and disagreed by 4.6%.

### 3.3. The Prescription and OTC Dispensing Patterns of Migraine Medications

To investigate the prescription patterns of migraine medications, pharmacists were asked about the most prescribed migraine medications by healthcare specialists and provided with four choices to choose from. The included drugs were triptans, opioids (both are prescription drugs), and NSAIDs and paracetamol (nonprescription drugs). Triptans were most prescribed for migraine patients, accounting for almost half of the pharmacists’ opinions (52.1%). NSAIDs such as ibuprofen were the second-most defined according to 26.0% of the study participants. In comparison, paracetamol was third (19.5%), followed by opioids, which were selected as the least prescribed drugs for migraine, among the given choices, as less than 1% of pharmacists’ opinions (Table 3).

Pharmacists were given a compilation of over-the-counter medications regarding the migraine medication most recommended by pharmacists. Triptans were the most recommended (37.2%), followed by NSAIDs such as ibuprofen (33.9%). Paracetamol was recommended by 23.3% of the pharmacists, and nonmedication therapy by 5.6% (Table 3).

### 3.4. Correlating Knowledge, Attitude, and Practice Scores to the Characteristics of Pharmacists

Results of the Mann-Whitney U test and Kruskal-Wallis H test showed no significant statistical difference between the community pharmacists’ average knowledge, attitude, and practice overall score and their demographic and practice variables (*p* > 0.05). Still, their overall scores significantly differed according to their gender (*p* = 0.04). Male pharmacists had higher overall scores than female pharmacists. These results are presented in Table 4.

## 4. Discussion

Community pharmacists are considered the first point of contact for the management of migraine patients. They also play an essential role in managing this condition [32,33]. However, the practice and knowledge of community pharmacists regarding migraine management can vary depending on the country and part where they practice and the specific laws and regulations governing migraine in that area. This study aims to assess the knowledge and practices of community pharmacists regarding migraine management. Our findings showed that most community pharmacists (96.7%) suggested OTC daily. Similarly, Salih and Abd found that most Iraqi pharmacists working in community pharmacies recommended OTC headache remedies at least once daily [34].

In this study, the most prescribed migraine treatments were triptans (52.2%), followed by NSAIDs (26.0%). Additionally, community pharmacists recommended prescription triptans (37.2%) as the first choice and OTC NSAIDs (33.9%) as the second choice. This contradicts Al-quliti et al.’s findings that most pharmacists suggest NSAIDs as a treatment for patients suffering from headaches [26]. Booth et al. found that most Western Australian community pharmacists (59.3%) would not recommend triptans when treating migraine [35]. Consistent with the literature [22,34], this study showed that most community pharmacists (86.0%) indicated that they feel that headache patients are an essential part of their community practice. In comparison, approximately 75.3% discuss non-drug treatment options with migraine patients, and 69.7% know when to make a physician referral.

Consistent with previous research [22,26], the findings showed that 9.3% of pharmacists disagreed that migraines are primarily neurological disorders, and 16.3% were neutral, despite the increasing knowledge of the pathophysiology of migraines [36]. Diamond found that many Health care professionals (HCPs) incorrectly attribute migraines to other issues, such as depression, anxiety, or stress [37]. This error, in turn, misdiagnoses and improperly treats the condition. Furthermore, it can cause patients to suffer from migraines for longer and further complicate matters.

Moreover, HCPs and patients may communicate poorly when such beliefs prevail, resulting in poor treatment outcomes [22]. Therefore, pharmacists must understand migraine as a neurobiological disease to help patients and HCPs select appropriate migraine treatments. Additionally, a continuing professional development (CPD) program should be developed to ensure that community pharmacists understand the pathophysiology of migraines.

Among the community pharmacists in this study, 33.9% recommended step care and nonspecific therapy, such as analgesics and NSAIDs, which remain the most common migraine treatment method in Saudi Arabia [16]. A cost-effective alternative to this approach is stratified care, in which the initial treatment is selected based on the patient’s treatment needs [38]. Previous research has supported this alternative approach [39,40,41]. Lipton et al. indicated that stratified care provides significantly better clinical outcomes as a treatment strategy than do step care strategies within or across attacks as measured by headache response and disability time [40]. Kosinski et al. reported that to assess headache-related morbidity and facilitate stratified care, a variety of validated tools have been developed, including the Migraine Disability Assessment Questionnaire (MIDAS) and the Headache Impact Test (HIT) [42]. In community pharmacies, these tools might screen patients complaining of headaches, direct them toward appropriate care, and track their progress over time.

Apart from migraines, a notable proportion of pharmacists’ responses were related to “sinus” headaches (28.3%). Previous studies have shown sinus headaches remain poorly defined, misunderstood, and often incorrectly associated with migraines despite being commonly diagnosed by general practitioners and self-diagnosing people [43,44]. Cady and coworkers indicated that the misdiagnosis of sinus headaches is widespread, with numerous instances of self-diagnosed diagnoses made by general practitioners turning out to be migraines [43]. As a result, pharmacists have the potential to assist patients in dispelling misconceptions surrounding sinus headaches [22].

Although most pharmacists advise patients to avoid excessive use of both OTC and prescription medications, medication overuse headaches still impact millions of individuals [45]. This finding indicates that the current efforts of pharmacists may not be sufficient [46]. To lessen medication-related headaches and decrease healthcare usage, pharmacists can proactively counsel patients on limiting the use of acute agents. Additionally, they can identify patients exceeding these limits and refer them for further assistance. According to Wenzel et al., pharmacists can reduce chronic daily headaches (CDH) related to medication overuse and healthcare resource consumption by proactively counseling patients to limit acute agents to no more than two days per week, which is a Consortium recommendation [22]. Almost 70% of community pharmacists feel confident in recognizing patients needing physician referrals. However, more research is required to determine the reasons behind such referrals and to establish criteria and guidelines for physician referrals, such as elevated MIDAS or HIT scores.

This study observed a significant difference between pharmacists’ gender and overall knowledge, attitude, and practice scores. Male pharmacists had more positive overall scores than female pharmacists. This finding broadly supports the work of Alzahrani et al. on smoking cessation, linking community pharmacists’ genders with providing smoking cessation activities [47]. This observation may reflect Saudi socio-cultural norms, discouraging female healthcare professionals from discussing or providing services to male patients. A recent study on Saudi female nurses in the Qassim part of Saudi Arabia reveals that more than 70% of them were unwilling to provide services to male patients. Most nurses also prefer working in female units, and 51% did not like night shifts [48].

The study’s outcomes imply a requirement for educational programs to enhance awareness and treatment of migraines, particularly involving community pharmacists who have the most frequent interactions with migraine patients [21]. Numerous resources, such as the Saudi guidelines for headache disorders and primary care networks, are accessible to back these endeavors. Multiple publications based on consensus are at hand to provide support, including the Saudi headache disorder guidelines and primary care network documents [16,49]. Providing urgent education and training for pharmacists is essential, particularly in areas such as differentiating levels of care, correctly utilizing OTC medications, addressing misunderstandings related to “sinus” headaches, and preventing medication overuse [9,22,39].

When interpreting the findings of a study, keeping in mind its limitations is essential. Except for one United States study, extensive validation studies have not been performed in a pharmacist population utilizing the study survey [22]. While the sample of community pharmacists in the current research matched the national labor force estimates on variables such as gender and foreign university graduation, the sample was younger than the national estimate [50]. Because of these variations and the inability to reach the required minimum sample size, the community pharmacists involved in this study do not fully represent all community pharmacists in the northwestern region of Saudi Arabia. It is important to note that the survey relied on self-reported data, which means some possible measurement errors. For example, Table 2 item 16, which examines pharmacists’ abilities to identify when to refer headache sufferers to a physician, is based on self-reports. This means pharmacists may have overestimated their success in migraine management services. It is worth noting that recall bias may have influenced some of the responses, which is common in studies examining past experiences. Lastly, it is crucial to remember that the study’s cross-sectional design does not permit any causal inferences. Despite these limitations, the study still provides valuable insights into pharmacists’ experiences and attitudes toward managing migraines in this particular context.

## 5. Conclusions

Interestingly, most community pharmacists recognize migraine as a primary brain disease, highlighting their knowledge of the condition. As members of healthcare who interact with migraine patients frequently, pharmacists are in a unique position to help promote better care. Nevertheless, it is important to highlight that not all pharmacists may be informed about the most recent guidelines for managing migraines. It’s worth noting that a significant number of pharmacists may not be up to date with the latest recommendations for migraine treatment. To address this, implementing enhanced training programs, continuing education initiatives, and fostering greater collaboration between pharmacists and healthcare professionals is important. With these measures in place, the efforts toward more effective migraine management in northwestern Saudi Arabia can be fruitful.

## Figures and Tables

**Table 1 pharmacy-11-00155-t001:** Characteristics of participating pharmacists (N = 215).

Characteristics	Categories	*n* (%)
Gender	Male	193 (89.8)
	Female	22 (10.2)
Age (Median (IQR))		31.0 (28–36)
Level of education	Bachelor’s Degree (BSc)	184 (85.6)
	PharmD	24 (11.2)
	Master’s degree (MSc)	5 (2.3)
	Doctorate Degree (PhD)	2 (0.9)
Years of experience as a pharmacist	Less than one year	3 (1.4)
	1–5 years	63 (29.3)
	6–10 years	70 (32.6)
	More than ten years	79 (36.7)
Source of the pharmacy degree	Saudi university	51 (23.7)
	Foreign university	164 (76.3)
Frequency of OTC headache product suggestion per day	1–5	180 (83.7)
	Six or more	28 (13.0)
	No suggestion	7 (3.3)
The most common headache conditions in your practice are 33.3% (one-third) or higher	Migraine	57 (26.5)
	Sinus headache	61 (28.3)
	Chronic headache	33 (15.3)
	Tension headache	40 (20.9)
	Do not know	24 (11.1)
Attending a course or educational event about headache/migraine management	No	97 (45.1)
	Yes	118 (54.9)
Frequency of referral to a physician for headache/migraine complaints per month	At least one	110 (51.1)
	At least two	48 (22.3)
	At least three	57 (26.5)
The primary basis for directing a patient to a physician	Infectiveness of the current treatment	62 (28.8)
	Quality of life	18 (8.4)
	Overuse of medications	38 (17.6)
	Suspicion of organic illness	97 (45.2)
Number of patients visiting your pharmacy suffering from migraines monthly	Less than 5	89 (41.4)
	5–10 patient	60 (27.9)
	11–15 patient	21 (9.8)
	More than 15	45 (20.9)

**Table 2 pharmacy-11-00155-t002:** Knowledge, attitude, and practice of pharmacists regarding migraine.

Statements	Strongly Disagree *n* (%)	Disagree *n* (%)	Neutral *n* (%)	Agree *n* (%)	Strongly Agree *n* (%)	Mean Score (SD)
Migraine is predominantly a brain disorder with a firmly established neurological foundation.	2 (0.9)	18 (8.4)	35 (16.3)	117 (54.4)	43 (20.0)	3.84 (0.87)
Individuals with migraines should initially consider using OTC medications and, if needed, explore prescription medications.	3 (1.4)	15 (7.0)	17 (7.9)	119 (55.3)	61 (28.4)	4.02 (0.88)
Patients who haven’t found relief with at least two other prescription medications should be considered for migraine-specific drugs (triptans).	2 (0.9)	10 (4.7)	48 (22.3)	124 (57.7)	31 (14.4)	3.80 (0.78)
Regular doses of NSAIDs are efficient for addressing acute migraine episodes.	4 (1.9)	41 (19.1)	52 (24.2)	92 (42.8)	26 (12.1)	3.44 (0.99)
The routine use of opioids and compounds containing barbiturates should be avoided to immediately treat symptoms linked to migraines.	3 (1.4)	15 (7.0)	41 (19.1)	100 (46.5)	56 (26.0)	3.89 (0.92)
Individuals with headaches constitute a significant portion of my pharmacy-related activities.	1 (0.5)	4 (1.9)	25 (11.6)	105 (48.8)	80 (37.2)	4.20 (0.75)
Ensuring patient contentment is a crucial factor in the treatment of headaches.	0 (0.0)	2 (0.9)	24 (11.2)	94 (43.7)	95 (44.2)	4.31 (0.70)
Abortive therapy treatment of symptoms associated with migraine should be used as early as possible during migraine management.	3 (1.4)	8 (3.7)	32 (14.9)	111 (51.6)	61 (28.4)	4.02 (0.84)
I consistently motivate headache patients to uphold a balanced and nutritious diet.	0 (0)	2 (0.9)	42 (19.5)	91 (42.3)	80 (37.2)	4.16 (0.76)
I consistently engage in conversations about non-medication therapies as an integral aspect of headache treatment.	0 (0)	16 (7.4)	37 (17.2)	97 (45.1)	65 (30.2)	3.98 (0.88)
I educate patients about the importance of avoiding excessive use of over-the-counter medications.	0 (0)	4 (1.9)	33 (15.3)	74 (34.4)	104 (48.4)	4.29 (0.79)
I teach patients about the necessity of preventing excessive usage of prescription medications.	1 (0.5)	9 (4.2)	40 (18.6)	80 (37.2)	85 (39.5)	4.11 (0.88)
Prior to recommending an OTC drug, I consistently inquired about the severity of the headache episode from the patient.	1 (0.5)	1 (0.5)	27 (12.6)	97 (45.1)	89 (41.4)	4.27 (0.73)
I have the ability to recognize patients who require migraine preventive medications.	0 (0)	7 (3.3)	69 (32.1)	117 (54.4)	22 (10.2)	3.72 (0.69)
I am knowledgeable about when to direct individuals experiencing headaches to a physician.	2 (0.9)	8 (3.7)	55 (25.6)	119 (55.3)	31 (14.4)	3.79 (0.77)

**Table 3 pharmacy-11-00155-t003:** The most prescribed and recommended medications used in migraine medications.

Statements	Categories	*n* (%)
What is the most prescribed medication used in migraine therapy?	triptans	112 (52.1)
	NSAIDs such as ibuprofen	56 (26.0)
	paracetamol	42 (19.5)
	Opioids	2 (0.9)
	Others	3 (1.4)
What is the most medication you recommend in migraine therapy?	triptans	80 (37.2)
	NSAIDs such as ibuprofen	73 (33.9)
	paracetamol	50 (23.3%)
	Non-medication therapy	12 (5.6)

**Table 4 pharmacy-11-00155-t004:** Comparison of Pharmacists’ knowledge and practice scores across characteristics of the pharmacists.

Variables	*n*	Overall Score Mean (SD)	*p*-Value	Effect Size
Age			0.07	
21–29	74	3.81 (0.43)		
30–39	117	3.93 (0.77)		
40–49	8	3.74 (0.18)		
50–60	3	4.12 (0.94)		
Gender			0.04 *	0.14
Male	193	3.87 (0.39)		
Female	22	3.71 (0.43)		
Level of education			0.22	
Bachelor’s Degree (BSc)	184	3.86 (0.40)		
PharmD	24	3.92 (0.37)		
Master’s degree (MSc)	5	4.2 (0.33)		
Doctorate Degree (PhD)	2	3.75 (0.88)		
Years of experience as a pharmacist			0.37	
Less than one year	3	3.97 (0.48)		
1–5 years	56	3.79 (0.41)		
6–10 years	70	3.90 (0.45)		
More than ten years	79	3.91 (0.32)		
Source of the pharmacy degree			0.22	
Saudi university	51	3.81 (0.41)		
Foreign university	164	3.89 (0.39)		
Frequency of OTC headache product suggestion per day			0.61	
1–5	180	3.85 (0.70)		
Six or more	28	3.97 (0.35)		
No suggestion	7	4.1 (0.27)		
The most common headache conditions that exist in your practice comprise one-third (33.3%) or more of your patients.			0.20	
Migraine	57	3.79 (0.40)		
Sinus headache	61	3.91 (0.35)		
Chronic headache	28	3.79 (0.41)		
Tension headache	45	3.89 (0.43)		
Do not know	24	4.00 (0.37)		
Attending a course or educational event about headache/migraine management			0.30	
No	97	3.83 (0.38)		
Yes	118	3.91 (0.41)		
Frequency of referral to a physician for headache/migraine complaints per month			0.12	
At least one	109	3.84 (0.38)		
At least two	48	3.85 (0.39)		
At least three	58	3.96 (0.43)		
The prevalent cause for directing a patient to a physician.			0.08	
Infectiveness of the current treatment	62	3.83 (0.39)		
Decreased quality of life	18	3.90 (0.35)		
Overuse of OTC and/or prescription medications	38	3.77 (0.36)		
Suspicion of organic illness	97	3.94 (0.41)		
Number of patients visiting your pharmacy suffering from migraines monthly			0.16	
Less than 5	88	3.85 (0.39)		
5–10 patient	60	3.82 (0.38)		
11–15 patient	21	3.92 (0.37)		
More than 15	46	3.97 (0.43)		

* Significant result (*p* < 0.05).

## Data Availability

Data are available by contacting the corresponding author upon reasonable request.

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
