# Peer review of "Migraine Management in Community Pharmacies: Knowledge, Attitude and Practice Patterns of Pharmacists in Saudi Arabia"

_pharmacy, 2023, doi:10.3390/pharmacy11050155_

Round 1

Reviewer 1 Report

Thank you for submitting this interesting article.  It is well written and of a research topic that is important to pharmacy practice.  There are some minor revisions that I could suggest that would make this article ready for publication.

Would the authors consider:

Using migraine instead of variations of the word in several places for consistency.  For example, the abstract has migraineurs which is a difficult word to understand and may not be in all dictionaries; and migration in the discussion. This would make the article less prone to misinterpretation.

Abstract line 16 - consider changing the sentence to "feel that migraine management is essential".

Abstract line 22 change p. value to (p-value=0.04)

Introduction line 32 in the sentence "The exact cause of is not fully understood", you need to add of what? or take out "of".

Materials and Methods line 93. Take out "from it" in the sentence ending "during the study was excluded from it."  Also the next sentences makes no sense and I feel it is a comment from another reviewer that was forgotten about and left.  Take these two sentences out or clarify what they are for.

Results Line 153.  Could the authors include a response rate if it is possible to calculate one.  You mention it in the discussion that you had a small response rate, but do not show what it is.

Results line 155 and Table 1 - could the age be changed to 1 decimal place.  Most people do not know their age to 2 decimal places.

Results line 210 - ibuprofen does not need to be capitalised, it is a generic name.

Results line 211 - you state that paracetamol was third (20.9%) prescribed drug, but Table 3 has 19.5%?  Which is correct?

The references have to be closely reviewed and revised.  Some references have journals in italics others do not, be consistent and to the editors preference (1,2,5,11, etc.) ; other references do not state the journal at all (3,4,18, etc.). Watch capitalisation of the journal titles.  Some journals are in long form (full name) others are in short form.  For references 21 and 42 write out the full authors Pharmacy Times (21) and Ministry of Health (42).  Also put the country that the Statistical Yearbook is for?

Finally, the findings that 25.6% of experienced pharmacists do not believe that migraine is a primarily neurological disorder is disturbing.  There should be a continuing professional development (CPD) programme developed to ensure that the pathophysiology of migraines in understood by community pharmacists.

The quality of English is the article is good, there are minor revisions that are needed, but overall it is easy to read.

Reviewer 2 Report

Many thanks to “Pharmacy” for letting me review this very interesting article. I enjoyed reading the manuscript. I have many comments and hope that they can contribute to the further improvement of the manuscript. I have inserted my comments directly into the word proof so that it is clearer to which parts of the manuscript they refer.

Moderate editing required.

Author Response

Dear Reviewer, 

Best Regards

Round 2

Reviewer 2 Report

Many thanks to PHARMACY for also allowing me to review the revised version of this interesting article. Many thanks also to the authors for the very carefully made revisions. I have only some minor comments, see the attached file.

Author Response

Dear Reviewer,

Thank you for taking the time to review our paper. We have made revisions based on your comments. Please see the attachment.

Best Regards,

Authors
